# Contrasting Autoimmune Comorbidities in Microscopic Colitis and Inflammatory Bowel Diseases

**DOI:** 10.3390/life13030652

**Published:** 2023-02-27

**Authors:** Istvan Fedor, Eva Zold, Zsolt Barta

**Affiliations:** 1Department of Public Health and Epidemiology, Faculty of Medicine, University of Debrecen, Kassai Street 26, 4012 Debrecen, Hungary; 2Department of Clinical Immunology, Faculty of Medicine, Institute of Internal Medicine, Doctoral School of Clinical Immunology and Allergology, University of Debrecen, Moricz Zs. Street 22, 4032 Debrecen, Hungary; 3GI Unit, Department of Infectology, Faculty of Medicine, Doctoral School of Clinical Immunology and Allergology, University of Debrecen, Bartok Bela Street 2-26, 4031 Debrecen, Hungary

**Keywords:** autoimmunity, inflammation, inflammatory bowel disease, microscopic colitis

## Abstract

Background: Inflammatory bowel diseases (Crohn’s disease and ulcerative colitis) and microscopic colitis (lymphocytic and collagenous colitis) are immune-mediated diseases of the gastrointestinal tract, with distinct pathophysiology. Objective: We sought to compare the prevalence of autoimmune diseases between microscopic colitis (MC) and inflammatory bowel diseases (IBDs) in our patient cohorts in their medical history. Methods: We collected data from 611 patients (508 with IBD, 103 with MC). We recorded cases of other autoimmune diseases. The screened documentation was written in the period between 2008 and 2022. We sought to determine whether colonic involvement had an impact on the prevalence of autoimmune diseases. Results: Ulcerative colitis patients and patients with colonic-predominant Crohn’s disease had a greater propensity for autoimmune conditions across the disease course than patients with ileal-predominant Crohn’s disease. Gluten-related disorders were more common in Crohn’s disease than in ulcerative colitis, and slightly more common than in microscopic colitis. In ulcerative colitis, 10 patients had non-differentiated collagenosis registered, which can later develop into a definite autoimmune disease. Conclusions: Predominantly colonic involvement can be a predisposing factor for developing additional autoimmune disorders in IBD. Ulcerative colitis patients may have laboratory markers of autoimmunity, without fulfilling the diagnostic criteria for definitive autoimmune disorders (non-differentiated collagenosis).

## 1. Introduction

Albeit both microscopic colitis (MC) and inflammatory bowel diseases (IBDs) are immune-mediated disorders predominantly affecting the gastrointestinal tract, there are delicate differences in their presentation [1,2,3]. Inflammatory bowel diseases are known to display extraintestinal manifestations (EIMs) during the disease course [4]. These extraintestinal findings are usually not found in microscopic colitis. On the other hand, microscopic colitis is mostly recognized for the frequent accompanying autoimmune diseases [1,5]. Although IBD shows a correlation with an increased tendency for other immune–inflammatory diseases, the level of association does not reach the magnitude seen in MC. Neither of these conditions (MC and IBD) are regarded as classical autoimmune conditions, as they lack specific autoantibodies, and the pathogenesis of IBD is characterized by a pathologic reaction to the commensal intestinal microbiome. The role of microbes in the pathogenesis of MC is indicated via the observation that fecal stream diversion with ileostomy contributes to symptom resolution and intestinal healing [6]. To further indicate the possible role of intestinal microbes, ileostomy patients frequently relapse, once intestinal continuity is restored [7]. Moreover, in refractory cases, fecal microbiota transplantation (FMBT) may offer benefits [8,9], though the careful selection of the donor sample is crucial [10]. Traditionally, these diseases display increasing prevalence in rapidly developing countries, through the incidence plateaued in developed, welfare societies. Thus, it is very likely to be multifactorial in etiology, and the genetic background is not sufficient in itself for developing these conditions. As all of these immune-mediated diseases of the gastrointestinal system are distinct—though they overlap in certain characteristics—we attempted to highlight the distinguishing features through our patients.

## 2. Patients and Methods

Our paper is an extension of two of our earlier datasets of patients with microscopic colitis (MC—103 patients) and inflammatory bowel disease (IBD—508 patients) [11,12]. In total, we included 611 patients in the current paper, within four groups. In the MC cohort, we had 103 patients, 67 females, 36 males with two different forms of MC (28 and 75 patients with lymphocytic and collagenous colitis, respectively). The IBD cohort consisted of 508 (303 with CD and 205 with UC) patients, 133 male and 170 female subjects with Crohn’s disease, and 89 male, and 116 female with ulcerative colitis. Apart from extending our registered data in the previous papers, we also aimed to compare different disease subgroups. For the basic characteristics of different subgroups, please refer to Table 1.

To determine whether a patient had an immune-mediated disease, we read through the available previous medical records from the year 2008 until 2022. The reason behind opting for this method was that healthcare administration systems are optimized for administrative and financial purposes, thus the patient history does not always correspond to the entries below the “Diagnosis” tabs. To overcome this, we screened the whole text of each visit, and only registered those cases wherein the history indicated an autoimmune disease.

Patient privacy was preserved by the use of anonymized data in this retrospective, non-interventional cohort study. The study protocol was approved by the local Ethics Committee which conformed to the provisions of the Declaration of Helsinki.

As a retrospective observational study, we first recorded patient data in an Excel spreadsheet (Excel 2019, Microsoft Corporation, Redmond, Washington, DC, USA). The extracted data were analyzed with Medcalc Software (Medcalc Software Ltd. Version 20.014, Ostend, Belgium). We aimed to include as many patients whose health records were accessible for reviewing. We were particularly interested in the prevalence of diseases of autoimmune origin (AI diseases) in our IBD cohort, and also contrasting these data with our MC cohorts. Comparing the proportions with AI diseases was performed via chi-squared test [13]. We also compared the average ages at diagnosis, as well as the different entities that affected patients in different life stages. For contrasting age at diagnosis, an independent sample two-tailed *t*-test was used.

## 3. Results

### 3.1. General Patient Characteristics of IBD and MC Cohorts

For the general characteristics of our patients, please refer to Table 1. The findings reflect the commonly reported patterns, though remarkably, microscopic colitis patients were younger than expected—especially those with lymphocytic colitis. We would also like to highlight that the male-to-female ratio was more balanced in collagenous rather than lymphocytic colitis. The latter displayed a strong female predominance, whereas both IBD entities were more balanced. The difference between autoimmune comorbidities is marked between the MC and IBD cohorts. Moreover, we found that the difference within disease subgroups did not differ markedly (lymphocytic colitis vs. collagenous colitis, and CD vs. UC). The disease entities showed distinct pattern in the average age of diagnosis. For the proportions of patients in different subgroups organized according to age of diagnosis, please see Table 2. 

### 3.2. Autoimmune Diseases in IBD and MC

In our patient cohorts, we found a roughly two-fold difference between disease conditions. 38.8% (40 patients out of 103) of all MC patients who had an AI-disease in their history; this ratio was 19.1% in all IBD cases (97 patients from 508 total). This difference was marked, with a 19.7% difference between IBD and MC cohorts, (95% CI: 10.1324–29.8622; *p* < 0.0001). As MC is confined to the colon (similar to ulcerative colitis—UC and the colonic subtype of Crohn’s disease: Montréal L2), we were interested in whether the “colonic predominant” phenotypes are more prone to develop other autoimmune diseases. Thereby, we compared the proportions of patients with accompanying autoimmune disorders in different Crohn’s disease subgroups (ileum and small bowel predominant, colonic predominant). For the results and possible implications, please refer to Table 3.

### 3.3. Colonic Involvement

Table 3, note that CD with predominantly colonic involvement displayed a higher proportion of accompanying autoimmune diseases. Differences between small intestinal predominant and colonic dominant Crohn’s disease reached a statistically significant level, whereas 29 patients in the small intestinal predominant group developed AI disease (corresponding to 14.2% of all patients with small-intestinal predominant Crohn’s), the ratio was 24.7% in the colonic-predominant subgroup (22 patients with L2 Crohn’s disease). This 10.5% difference between the groups was significant (*p* = 0.0297).

By incorporating data from UC, 23.1% of patients with pure colonic IBD had other immune-mediated inflammatory disorders in their history (68 patients out of 294—UC and CD L2 combined), whereas CD patients with predominantly small-intestinal involvement had autoimmune disorders in 29 cases out of 204 patients total (14.2%). The difference reached a statistically significant level.

Patients in the UC cohort had a similar tendency for developing AI disorders to colonic Crohn’s disease patients. A total of 22 patients with L2 Crohn and 46 patients with UC had one or more additional AI diseases in their histories (24.7% and 22.4%, respectively). This marginal difference (2.3%) yielded a non-significant result (*p* = 0.6678). Therefore, purely colonic Crohn’s and UC patients had a similar risk for other autoimmune diseases.

As MC shares the characteristic of pure colonic involvement with UC and L2 Crohn’s disease, we also assessed differences in comorbid AI diseases between these states. In MC (total patients: 103, patients with autoimmune diseases: 40 (38.8%)) we saw a marked tendency for other autoimmune diseases in comparison with colonic IBD (difference: 15.7%, 95% CI: 5.4351–26.2999 *p* = 0.0021—for the data, please refer to Table 3 and Table 5). Thus, autoimmune disorders are much more prevalent in MC.

In Table 4 it is apparent that Crohn’s disease patients were less likely to develop other diseases of autoimmunity, though the difference between the groups did not reach a statistically significant level. While 16.8% of all patients with Crohn’s disease had other autoimmune diseases in their medical history, this ratio was 22.4% in UC (51 patients out of 303 in CD and 46 patients out of 205 in UC). The 5.6% difference between the groups is not significant (*p* = 0.1153). This finding is contrary to previous reports, and we believe it may both reflect regional differences as well as a possible bias due to the small number of cases.

### 3.4. Undifferentiated Connective Tissue Disease (UCTD) in Ulcerative Colitis

In our cohort, patients with UC were more prone to develop the condition of undifferentiated connective tissue disease (UCTD). While 10 patients (roughly 4.9%) with UC had UCTD in their disease histories, this was not as high in Crohn’s patients (four patients, approximately 1.3%). It is worth noting that the UC in this regard was comparable to microscopic colitis (please refer to Table 5). These patients had autoantibodies characteristic of manifest autoimmune diseases, but they did not meet the full diagnostic criteria for definite disease. The conversion of UCTD to overt autoimmune disease is variable. According to a 2009 publication by Bodolay et al., the conversion rate is roughly 30 to 40% in a five-year course [14]. Most patients do not progress into AI diseases within five years of follow-up, and the prognosis of later autoimmune diseases is more favorable with preceding UCTD. All of the patients who had UCTD in their medical records were female.

### 3.5. Gluten-Related Disorders in IBD

Gluten-related disorders were more prevalent in Crohn’s disease than in UC. In Crohn’s disease, there were 12 cases (approximately 4% of total patients) with accompanying celiac disease (gluten-sensitive enteropathy—GSE), whereas this disease affected two (1%) patients with UC. The ratio of patients with GSE was very similar in MC to that seen in CD: 3.9% of all patients had GSE in their disease histories.

A total of two patients with CD were registered with non-celiac gluten sensitivity—NCGS—while none of the UC patients had this in their medical records. Another gluten-related disorder, dermatitis herpetiformis—DH—was present in two of CD and one of UC cases (approximately 0.7% and 0.5%, respectively). Remarkably, none of the patients in the MC cohort had dermatitis herpetiformis.

In the CD cohort, patients with upper-intestinal involvement—L4 subtype—had a high prevalence of accompanying AI diseases. A total of four out of six patients with upper intestinal lesions had comorbid AI disease in their histories, although three of these cases (75%) were celiac disease—GSE. The two disease entities have distinct histopathology and biopsy-confirmed alterations, characteristic of GSE in these cases.

### 3.6. Rheumatoid Arthritis (RA) in IBD

IBD patients with RA pose a differential diagnostic challenge if they also have arthropathy. Peripheral arthritis may be an extraintestinal finding of their IBD. A total of 5 of CD patients (approximately 1.7%) and 11 of UC patients (approximately 5.4%) had RA in their medical records registered, whereas this ratio reached 6.9% in cases with MC (seven patients, LC and CC combined). Moreover, RA presented the most marked difference between IBD subgroups: the difference between CD and UC proved to be significant (*p* = 0.0187). Thereby, one can conclude that RA seems to be more prevalent in UC than in CD. In cases of other immune-mediated diseases, the difference between Crohn’s disease and UC was not as marked. We believe our study was underpowered in revealing any additional differences.

### 3.7. Hepatobiliary Autoimmune Diseases

It is noteworthy to add that we did not classify primary sclerosing cholangitis as a distinct autoimmune disease entity, but rather opted for recognizing the condition as an extra-intestinal manifestation of IBD [15]. PSC without underlying IBD is rare. Generally, PSC does not develop in MC, and it is mostly regarded as an EIM of IBD. There is scarce evidence in the literature for MC patients with PSC [16]. In our cohort, there were no patients with PSC in the MC cohort.

In our IBD cohort, only three patients developed either primary biliary cholangitis or autoimmune hepatitis (PBC or AIH); thus, not even 1% of all IBD patients had a hepatobiliary autoimmune disorder in their histories. In the microscopic colitis cohort—even though 104 patients’ data were screened, there was only one case of AIH and no cases of PBC.

### 3.8. Autoimmune Diseases of the Thyroid

Hashimoto thyroiditis was prevalent in both CD and UC. Thyroid-related problems are not uncommon in the population, and we found Hashimoto thyroiditis to be the most common in our MC cohort as well. A total of 15 patients from our MC cohort had thyroid-related abnormality—14 (37.5%) of them had Hashimoto thyroiditis in their medical records. The ratio in IBD was much less pronounced. 13 patients with CD (4.3%) and 10 with UC (4.9%) had thyroiditis of immune-mediated origin in their medical histories.

## 4. Discussion

Our paper has its limitations, being a cross-sectional observational retrospective study from available patient data [17,18]. We also did not have a healthy control group; therefore, we were only able to compare disease subgroups with the data from other publications. Nonetheless, we believe the paper has interesting findings regarding autoimmune comorbidities in IBD and their relationship to colonic involvement. 

For a general comparison of microscopic colitis and inflammatory bowel diseases, we included Table 6 for a brief cardinal overview of these entities. Our findings highlight that IBD and MC patients greatly differ in their risks for developing autoimmune disorders during their disease course. MC patients had a two-fold increase in the prevalence of other autoimmune conditions compared to our IBD cohort. However, one should interpret this seemingly large gap between the groups with caution. While it is true that most autoimmune diseases manifest in the young adults or middle-aged population, our IBD cohort was relatively younger than patients with MC (for the age of diagnosis, please refer to Table 2 and Figure 1). As autoimmune disorders may develop later in life, it would be more plausible to compare lifetime risks.

We should emphasize that we only assessed the prevalence of autoimmune disorders recorded in the history of patients. We did not assess—in a broader sense—all immune-mediated inflammatory disorders. Other authors (Convay et al. from Boston, US) reported total immune-mediated diseases; thus, they reported cases of asthma or skin disorders of atopic origin as well [19]. Remarkably, they found a much higher prevalence for psoriasis, whereas this was rare among our patients in IBD, comparable to otherwise healthy populations. Nonetheless, the prevalence of psoriasis in patients with IBD is generally higher than in the healthy population (please see the meta-analysis and systematic review by Alinaghi et al. [20]); therefore, it is possible that we had an underreport in this regard. We would also like to point out that not all authors reported an increased prevalence of psoriasis in IBD, though it seems that IBD predisposes the development of psoriatic arthritis [20,21]. Furthermore, certain biologic therapies (Secukinumab, an anti-IL17 monoclonal antibody) utilized in psoriasis may predispose patients to a subsequent development of IBD [22,23,24]. As this agent was not used in the management of any of our patients, we can safely rule out this possibility. Generally, the development of IBD in patients receiving Secukinumab is unlikely, and the agent was found to be safe (please refer to the meta-analysis by Schreiber et al. [22]). In connection with psoriasis, we found zero cases in our microscopic colitis cohort. We would like to emphasize the limited sample size as a possible explanation. We sincerely believe that, with the inclusion of larger samples, sporadic cases of psoriasis would have been registered.

In the IBD cohort, patients with the colon-predominant disease were more prone to develop other immune-mediated diseases. Some authors regard ileum-predominant and colonic-predominant CD to be distinct entities [25]. Moreover, the colonic-predominant types may display inefficiency, and loss of response to anti-TNF therapies [26].

Autoimmune conditions that differed significantly between Crohn’s disease and ulcerative colitis were rheumatoid arthritis (RA) and undifferentiated connective tissue disease (UCTD). Contrary to our expectations, patients with ulcerative colitis had a greater prevalence of RA in their histories [27,28]. This conflicts with the data presented in the systematic review and meta-analysis, conducted by Chen and coworkers (BMC Gastroenterology 2020 [29]). As they included eight studies in their meta-analysis, and indeed, most of the studies reported RA to be more prevalent in Crohn’s disease, we believe that perhaps our sample was too small in size to observe this phenomenon. Another possible explanation for our findings is regional differences. One should bear in mind that the previously published data seems to favor the concept, of CD patients being more prone to develop additional immune-mediated diseases, but some authors described very similar representations. As our cohorts did not differ significantly in terms of autoimmune comorbidities, we would not like to draw any conclusions from our findings. It is possible that, regionally, there is no marked difference in the presence of autoimmune diseases between different IBD cohorts.

We found that gluten-related comorbidities were more common in CD than in UC. As a side note, gluten (and casein)-free diets are gaining popularity as an ancillary lifestyle intervention in immune-mediated inflammatory disorders [30]. Nonetheless, the evidence is vague and weak; therefore, this approach cannot be recommended generally [31]. Please note that GSE (gluten-sensitive enteropathy, celiac disease) and dermatitis herpetiformis (DH) are indeed conditional autoimmune comorbidities with sensitive and specific autoantibodies, whereas non-celiac gluten sensitivity (NCGS) rarely displays any alterations in laboratory studies. Despite the non-celiac gluten sensitivity (NCGS) not being able to be classified as a classical autoimmune disorder, we included the registered cases, due to their relationship to gluten exposure [32,33]. Nonetheless, the pathophysiology behind NCGS is not well understood. The condition in itself is likely to be heterogeneous in origin. In certain patients, there could be alterations in immune regulation [33,34].

The low overall prevalence of hepatobiliary autoimmune diseases (AIH and PBC) poses the question of whether IBD or MC truly enhances the risk of developing these autoimmune conditions. We sincerely believe that this is the case: the overall prevalence of PBC and AIH is extremely low in the general population, and yet we were able to register four cases in the entire cohort (611 patients total, 508 patients with IBD, 103 patients with MC, four cases of AIH and PBC combined) [35,36].

IBD displayed showed a mildly increased prevalence of Hashimoto thyroiditis compared to the general population. The estimated prevalence of Hashimoto thyroiditis in the general population is 0.8–1%. We would like to mention that thyroid-related abnormalities are not uncommon in the otherwise healthy population, and the incidence of these seems to be on the rise. Therefore, we cannot rule out that environmental and epigenetic factors also contribute to thyroid–gland disorders, as well as to diseases of immune-mediated origin.

We would like to propose the concept of intestinal barrier dysfunction and “leaky gut” as a predisposing factor for developing immune-mediated inflammatory conditions. Even though it has been long known that these phenomena may be attenuated via additional glutamine supplementation, there is no recommendation for the use of this amino acid. According to our experiences in practice, patients do report increased energy and mood (thus, subjective quality of life) when administered glutamine supplementation. The literature nonetheless is controversial, on whether it truly offers benefits. Our view on the subject is that it is a plausible practice, with a sound physiologic background. Glutamine is not only a contributor to enterocyte proliferation (and thus the healing of intestinal lining) but it can also enhance tight-junction functions and reins pro-inflammatory signaling pathways [37]. While it is known that the incidence of newly diagnosed autoimmune disorders in IBD exceeds the numbers seen in healthy control populations, one may propose the idea that this risk can be decreased with adequate intestinal lining maintenance. Furthermore, prospective studies would provide greater insight into whether glutamine supplementation truly protects against immune-mediated inflammatory disorder development during a longer period. Even though a meta-analysis by Severo and colleagues in 2021 and a Cochrane review by Akobeng in 2016 found insufficient evidence that glutamine may have ameliorating effects in IBD, one cannot rule out the possible beneficial influences on disease course and life quality [38,39].

As there is evidence that intestinal barrier dysfunction and dysbiosis of microbiota may enhance the risk for autoimmune diseases, we sincerely believe that every factor contributing to healthy gut lining may be protective against the subsequent development of these conditions [40]. Certain factors that may disrupt the homeostasis of the gut microbiome are likely to contribute to both immune-mediated inflammatory diseases and leaky gut syndrome [41,42,43,44]. Even though the mechanisms are not entirely clear, excess, uncontrolled psychological stress was shown to predispose patients to a subsequent development of systemic immune diseases and IBD as well [45,46,47]. Moreover, the clinical course of IBD often reflects the psychological state of patients; therefore, it seems plausible that, in certain circumstances, psychological interventions (e.g., cognitive behavioral therapy—CBD) may provide benefits for patients with IBD [48]. Nonetheless, the consensus is not well established, and the evidence is vague, though these interventions have little risk. A Cochrane systematic review and meta-analysis found psychotherapies to be ineffective [49]. It is unclear why chronic stress exposure (thus, state of hypercortisolism) predisposes patients to immune-mediated inflammatory disorders, whereas the efficient therapies for these conditions generally contain (at least temporary) administration of exogenous glucocorticoids.

Another observation regarding glucocorticoids according to which current steroid use may pose a somewhat increased risk for the development of autoimmune diseases also sounds counterintuitive but is described nonetheless [50]. On the other hand, aminosalicylates may protect against these conditions, and thus can be recommended for indefinite use in patients with inflammatory boswel diseases.

## 5. Conclusions

Though there are overlaps in the presentation of microscopic colitis and inflammatory bowel diseases, they differ in fundamental characteristics. One of the key distinguishing features is the marked increase in autoimmune diseases in microscopic colitis. Moreover, patients with inflammatory bowel disease confined to the colonic segments displayed a higher prevalence of diseases of autoimmune origin. A substantial proportion of both ulcerative colitis and microscopic colitis patients had evidence of the presence of autoantibodies, without fulfilling the diagnostic criteria for overt autoimmune diseases (undifferentiated connective tissue disease). Crohn’s disease patients were more prone to develop gluten-related disorders, whereas it was not as prevalent in microscopic colitis as expected. As autoimmune diseases were more common in ulcerative colitis—as opposed to Crohn’s disease—there may be a regional characteristic in IBD, as other authors reported AI diseases to be more prevalent in CD. Though rheumatoid arthritis (RA) is regarded to be more frequent in CD, we saw a higher number of cases in our UC cohort, in a similar ratio to that seen in MC. 

## Figures and Tables

**Figure 1 life-13-00652-f001:**
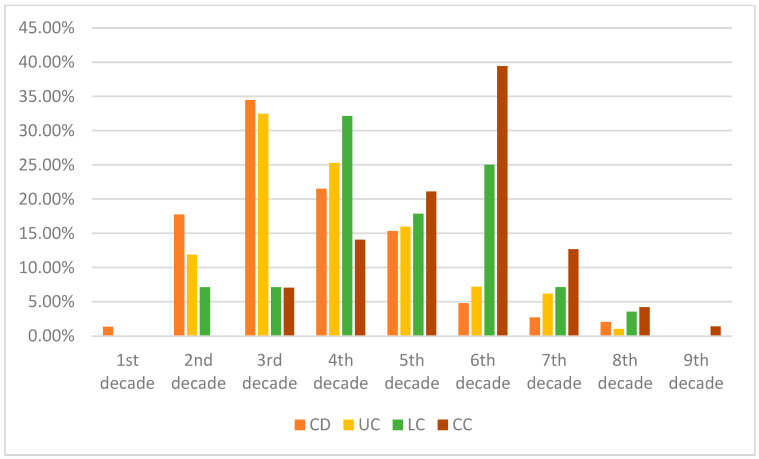
Comparison of average age at diagnosis of our patient cohorts—in the percentage of patients. It is worth noting that MC in the first two–three decades of life is extremely rare, though there is a possibility of under-report. It is worth noting that, in earlier life decades, Crohn’s disease had the highest diagnosis rate, whereas the incidence of UC is typically at a later age. Both subtypes of MC had a higher age of diagnosis.

**Table 1 life-13-00652-t001:** General patient characteristics in microscopic colitis and inflammatory bowel diseases.

	Microscopic Colitis		Inflammatory Bowel Diseases	
	Lymphocytic Colitis	Collagenous Colitis	Crohn’s Disease	Ulcerative Colitis
number of patients	103		508	
	28 (27.2%)	75 (72.8%)	303	205
sex:	5 male (18%); 23 female (82%) *p* < 0.0001	31 male (41.3%); 44 (58.7%) female *p* = 0.0337	133 male (43.9%); 170 (56.1%) female *p* = 0.0027	89 male (43.4%), 116 (56.6%) female *p* = 0.0076
age of diagnosis (years)	44.5 ± 5.3	51.9 ± 12.8	32.4 ± 12.2	35 ± 13.8
	Difference: 7.4 years (95% CI: 2.4383% to 12.3617%; *p* = 0.0038)		Difference: 2.6 years (95% CI: 0.3135% to 4.8865%; *p* = 0.0259)	
AI diseases and intergroup difference	10 (36%)	30 (40%)	51 (16.8%)	46 (22.4%)
	difference: 4 % (95% CI: -17.1256 % to 22.8420 %; *p* = 0.7124)		difference: 5.6 % (95 % CI: -1.3328% to 12.8528 %; *p* = 0.1153)	
AI diseases total	40 (38.8%)		97 (19.1%)	

**Table 2 life-13-00652-t002:** Age of diagnosis of Crohn’s disease (CD), ulcerative colitis (UC), lymphocytic (LC), and collagenous colitis (CC).

Age of Diagnosis	CD	UC	LC	CC
1st decade	1.36%	0%	0.00%	0.00%
2nd decade	17.75%	11.86%	7.14%	0.00%
3rd decade	34.47%	32.47%	7.14%	7.04%
4th decade	21.50%	25.26%	32.14%	14.08%
5th decade	15.36%	15.98%	17.86%	21.13%
6th decade	4.78%	7.22%	25.00%	39.44%
7th decade	2.73%	6.19%	7.14%	12.68%
8th decade	2.05%	1.03%	3.57%	4.23%
9th decade	0.00%	0%	0.00%	1.41%

It is worth noting that in the first two decades of life, microscopic colitides was rarely diagnosed in our cohort. We report the numbers in two decimal numbers so as to reflect the prevalence more accurately.

**Table 3 life-13-00652-t003:** Involved bowel segments and presence of autoimmune diseases in Crohn’s disease and ulcerative colitis patients.

Involved Bowel Segment	Small Intestinal Predominant Crohn (L1, L3, L4)	Colonic Crohn (L2)	Ulcerative Colitis
Autoimmune diseases	29 (204 patients total; 14.2%)	22 (89 patients total; 24.7%)	46 (205 patients total; 22.4%)
		Colonic IBD (CD L2 and UC combined): 68 (294 patients total; 23.1%)	
Differences between subgroups of patients in autoimmune diseases			
Difference between small intestinal and colonic Crohn’s disease: 10.5%; *p* = 0.0297		Difference between UC and small intestinal predominant Crohn’s disease: 8.2%; *p* = 0.0325	
Difference between small intestinal predominant Crohn’s disease and colonic IBD: 8.9%; *p* = 0.0135			

Note that patients with colonic Crohn’s (L2) and ulcerative colitis had greater prevalence for other autoimmune disorders than patients with predominantly small intestinal involvement.

**Table 4 life-13-00652-t004:** Comorbid conditions in inflammatory bowel diseases.

Autoimmune Diseases Total:	Crohn’s	Ulcerative Colitis	Difference (*p*-Values)
	51 (16.8%)	46 (22.4%)	5.6% (*p* = 0.1153)
Rheumatoid arthritis (RA)	5 (1.66%)	11 (5.37%)	3.72% (*p* = 0.0187)
Thyroiditis	13 (4.29%)	10 (4.88%)	0.59% (*p* = 0.7539)
Gluten-sensitive enteropathy (celiac disease)	12 (3.96%)	2 (0.98%)	2.98% (*p* = 0.0444)
Non-celiac gluten sensitivity (NCGS)	2 (0.66%)	0	0.66% (*p*= 0.2443)
Dermatitis Herpetiformis (DH)	2 (0.66%)	1 (0.49%)	0.17% (*p* = 0.8065)
Systemic Sclerosis	2 (0.66%)	0	0.66% (*p*= 0.2443)
Sjögren’s syndrome	4 (1.32%)	3 (1.46%)	0.14% (*p* = 0.8944)
Antiphospholipid syndrome (APS)	2 (0.66%)	2 (0.98%)	0.32% (*p* = 0.6895)
Addison’s disease	2 (0.66%)	1 (0.49%)	0.17% (*p* = 0.8065)
Multiple sclerosis (MS)	1 (0.33%)	0	0.33% (*p* = 0.4108)
Polymyositis	2 (0.66%)	0	0.66% (*p* = 0.2443)
Ankylosing spondylitis (AS)	1 (0.33%)	2 (0.98%)	0.65% (*p* = 0.3494)
Vitiligo	2 (0.66%)	0	0.66% (*p* = 0.2443)
Psoriasis	2 (0.66%)	3 (1.46%)	0.8% (*p* = 0.3703)
Primary biliary cholangitis (PBC)	1 (0.33%)	0	0.33% (*p* = 0.4108)
Autoimmune hepatitis (AIH)	0	2 (0.98%)	0.98% (*p* = 0.0845)
Undifferentiated connective tissue disease (UCTD)	4 (1.32%)	10 (4.88%)	3.56% (*p* = 0.0163)

Only rheumatoid arthritis and gluten-sensitive enteropathy differed significantly between groups. Even though NCGS is technically not a classical autoimmune disorder, we included the two cases, as a gluten-related entity, manifesting with symptoms. Moreover, whereas UCTD does not fulfill the diagnostic criteria for “overt” autoimmune disease, we also included it in the table, as there is evidence for autoimmunity in laboratory studies. Again, due to the small numbers, we opted for displaying results in two decimal places.

**Table 5 life-13-00652-t005:** Autoimmune conditions in microscopic colitis.

Lymphocytic Colitis	Collagenous Colitis
10 (36%)	30 (40%)
In total: 39% of all patients; difference between groups: 4%, *p* = 0.7124 All autoimmune disorders in both groups combined:	
Hashimoto thyroiditis	14 (13.59%)
Rheumatoid arthritis (RA)	7 (6.79%)
Sjögren’s syndrome	7 (6.79%)
Undifferentiated connective tissue disease (UCTD)	5 (4.85%)
Gluten Sensitive Enteropathy (celiac disease, GSE)	4 (3.88%)
Systemic Lupus Erythematosus (SLE)	4 (3.88%)
Mixed connective tissue diseases (MCTD)	1 (0.97%)
Ankylosing spondylitis (AS)	1 (0.97%)
Graves–Basedow thyroiditis	1 (0.97%)
Autoimmune hepatitis (AIH)	1 (0.97%)

**Table 6 life-13-00652-t006:** Summary of the differences between MC and IBD—from the clinician’s perspective.

	Microscopic Colitis	Inflammatory Bowel Disease
Typical age of presentation	Above 50 years of age, most commonly above 65 years.	Usually in the first three decades of life, particularly Crohn’s disease.
Presence of autoimmune comorbidities	Rather common (affecting more than one-third of patients). Almost exclusively diagnosed before bowel disease.	More common than in the general population, but roughly half of that is seen in microscopic colitis. They can present before or roughly at the same time as bowel symptoms.
Sex ratios	Females outnumber male cases by 2.5×.	More balanced ratios, with mild female predominance.
Affected bowel segments	Purely colonic, without lesions in upper intestinal segments.	Can present anywhere along the GI tract (Crohn’s disease), and UC may have terminal ileal inflammation (backwash ileitis).
Disease course	Usually benign, no risk of malignant transformation, very low risk for complications and strictures–scar tissue.	Greatly depends on management. Increased tendency for colorectal carcinomas, risk (esp. CD) for strictures, and even perforation.
Effective management	The vast majority of cases respond to budesonide (topical corticosteroid).	Various agents are shown to be effective. Both systemic and topical corticosteroids, a range of monoclonal antibodies, aminosalicylates, and other immune-modulatory agents.
Clinical symptoms	Watery diarrhea without bleeding, though the absence of diarrhea or reduced bowel motility does not rule out the disease.Symptoms frequently during early dawn hours.	Crohn’s disease may present late, only when there are strictures and hindered GI transit, whereas UC is usually detected earlier and more likely to have blood in the stool. Symptoms are random throughout the day and night.

## Data Availability

The data that support the findings of this study are available from the corresponding author (I.F.) upon reasonable request.

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
