# Peer review of "Contrasting Autoimmune Comorbidities in Microscopic Colitis and Inflammatory Bowel Diseases"

_life, 2023, doi:10.3390/life13030652_

Round 1

Reviewer 1 Report

The introduction is very short, it has only one paragraph. It should be expanded. The table is totally defaced. It is hard to read the data. I suggest redoing it. The methodology is also very short. It should be described in more detail. As the article would be published in an online version, therefore, colors are not a problem at least for the online version, I suggest that figure 1 should be in color. Table 3 should also be redone. I can not read properly. Better format and send again. Same problem with table 6. In the text, you also state that “it is apparent, that Crohn’s disease patients were less likely to develop other diseases of autoimmunity, although the difference between the groups did not reach a statistically significant level.” However, you leave this information a little orphaned from further explanation. Nothing is raised about this claim, not even a hypothesis. The discussion is a bit disjointed and poorly written. Needs to improve a lot. As well as the conclusion, which is unclear.

Author Response

Dear reviewer 1,

First of all, we would like to express our gratitude for your effort and time in reviewing our paper. We are also grateful, that you have made valuable suggestions for us to improve upon the former version of our manuscript.

Following your precious remarks, we included additional information on the patients and methods section.

Your suggestion about the figure is well justified, as the paper will be available online. We made minor improvements to Tables 1-2 and 6 to make them more easily interpretable.

We also included a proposal of plausible explanations for our findings in the frequency of autoimmune disorders in Crohn’s disease and ulcerative colitis. We also added a brief conclusions section and made adjustments to the discussion to make it more coherent and readable.

Again we are thankful for the effort and suggestions, and we sincerely hope, that the updated version of the paper is an improvement over the previous submission.

Faithfully yours

The Authors

Reviewer 2 Report

-        “Inflammatory bowel diseases are known to 52 display extraintestinal manifestations (EIM) during the disease course.”

Add a reference, for example “The gut and the inflammatory bowel diseases inside-out: extra-intestinal manifestations. Minerva Gastroenterol Dietol. 2019 Dec;65(4):309-318. doi: 10.23736/S1121-421X.19.02577-7. Epub 2019 Apr 16. PMID: 30994321.”

-        “On the other hand, microscopic colitis is mostly recognized for the frequent accompanying autoimmune diseases.”

Add a reference

-        “The role of microbes in the pathogenesis of MC is indicated via the observation, that certain surgical procedures contribute to symptom resolution and intestinal healing, and patients relapse, once intestinal continuity is restored.”

Please, be more specific

-        “As all of these immune-mediated diseases of the 66 gastrointestinal system s”

Re-check this sentence

-        Re-check structure of Table 1, Table 3

-        “highlight, that”

Be careful with the commas in the text

-        “14.21%” “24.7%”

Report only 1 digit after the point in the whole text

-        Explain all the abbreviations in Table 4

-        Is NCGS and AI ??

-        Add a conclusion

Author Response

Dear Reviewer 2,

First and foremost, we would like to thank you for your effort and time in reviewing our paper. We are also grateful, that you have made great suggestions for the improvement of our submission.

As suggested, we included a reference to extraintestinal manifestations in IBD (PMID: 30994321). Moreover, we added new references throughout different sections of the text.

Our brief overview of the possible implications of intestinal microbes in MC is more elaborated, now it includes the typically performed surgery and also the possibility of fecal microbiota transplantation.

Regarding your suggestion on reporting results in only one decimal: whenever the differences and ratios are minor, we rather opted for two decimals in the tables. However, we revised and rounded several percentages in the main text, as you suggested. I hope this approach (approximating in the main text, while displaying the exact results in tables) is both justifiable and reasonable.

Indeed, you are correct, that we should have listed abbreviations in table 4. Thereby, in the revision, we opted for the full name of different entities, to make it easier to follow the data.

Even though NCGS is not classified as a classic autoimmune disorder, we included the two cases due to the following reason: it is a condition that requires environmental exposure to gluten, and, even though the exact pathophysiology is not yet elucidated, in certain cases immune-mediated pathogenesis might be present (Cárdenas-Torres et al., 2021; Sharma et al., 2020).

Again we are most grateful for your thorough review and comments on the paper. We hope, that you find the implemented changes contributed to a better quality of the manuscript.

Sincerely yours

The Authors

Cárdenas-Torres, F. I., Cabrera-Chávez, F., Figueroa-Salcido, O. G., & Ontiveros, N. (2021). Non-celiac gluten sensitivity: An update. In Medicina (Lithuania) (Vol. 57, Issue 6). MDPI AG. https://doi.org/10.3390/medicina57060526

Sharma, N., Bhatia, S., Chunduri, V., Kaur, S., Sharma, S., Kapoor, P., Kumari, A., & Garg, M. (2020). Pathogenesis of Celiac Disease and Other Gluten Related Disorders in Wheat and Strategies for Mitigating Them. In Frontiers in Nutrition (Vol. 7). Frontiers Media S.A. https://doi.org/10.3389/fnut.2020.00006

Reviewer 3 Report

1. Please check grammar errors:

Line 34, line 77, line 83, line 121, line 297, line 301,

2. Please be more specific for your statistical analysis in the methods.

Author Response

Dear Reviewer 3,

First of all, we would like to thank you for reviewing and commenting on our paper. We found your suggestions well justified and thus attempted to overcome the raised issues in the revised version of the paper.

We included a brief extension on the methods section, as well as corrected several grammar mistakes (especially the use of proper prepositions).

We are most grateful for your suggestions and sincerely hope, that with the implemented adjustments, the paper improved in quality.

Faithfully yours,

The Authors

Round 2

Reviewer 1 Report

The authors have made significant improvements to the manuscript. The quality is much higher. The tables are readable, the presentation of results, material and methods, discussion and conclusion have also improved a lot. My decision is to approve the manuscript for publication as an article.